# Characterization of Second-Generation Snacks Manufactured from Andean Tubers and Tuberous Root Flours

**DOI:** 10.3390/foods13010051

**Published:** 2023-12-22

**Authors:** Liliana Acurio, Diego Salazar, Bagner Castillo, Cristian Santiana, Javier Martínez-Monzó, Marta Igual

**Affiliations:** 1Department of Science and Engineering in Food and Biotechnology, Technical University of Ambato, Av. Los Chasquis & Río Payamino, Ambato 180150, Ecuador; dm.salazar@uta.edu.ec (D.S.); bagner_2295@hotmail.com (B.C.); 2i-Food Group, Instituto Universitario de Ingeniería de Alimentos-FoodUPV, Universitat Politècnica de València, Camino de Vera s/n, 46021 Valencia, Spain; xmartine@tal.upv.es; 3Facultad de Ciencias Pecuarias, Escuela Superior Politécnica de Chimborazo (ESPOCH), Panamericana Sur Km 1 1/2, Riobamba 060155, Ecuador; cristian.santiana@espoch.edu.ec

**Keywords:** *Arracacia xanthorrhiza*, *Canna indica*, *Colocasia esculenta*, *Ipomoea batatas*, *Oxalis tuberosa*, *Tropaeolum tuberosum*

## Abstract

Andean roots, such as zanahoria blanca, achira, papa China, camote, oca, and mashua, contain high amounts of dietary fiber, vitamins, minerals, and fructo-oligosaccharides. This study aimed to demonstrate the possibility of obtaining healthy second-generation (2G) snacks (products obtained from the immediate expansion of the mixture at the exit of the extruder die) using these roots as raw materials. Corn grits were mixed with Andean root flour in a proportion of 80:20, and a Brabender laboratory extruder was used to obtain the 2G snacks. The addition of root flour increased the water content, water activity, sectional expansion index, hygroscopicity, bulk density, and water absorption index but decreased the porosity. However, all 2G snacks manufactured with Andean root flour showed better characteristics than did the control (made with corn grits) in texture (softer in the first bite and pleasant crispness) and optical properties (more intense and saturated colors). The developed snacks could be considered functional foods due to the high amount of carotenoids and phenolic compounds they exhibit after the addition of Andean root flours. The composition of raw roots, specifically the starch, fiber, and protein content, had the most impact on snack properties due to their gelatinization or denaturalization.

## 1. Introduction

South America is one of the most megadiverse regions worldwide due to its biological variety, cultural richness, and economic potential. Contrary to popular belief, it has low food production and high malnutrition rates [1]. Developing nutritious foods from local crops poses a research challenge. The local crops comprise Andean roots, such as zanahoria blanca (*Arracacia xanthorrhiza* Bancr.), achira (*Canna indica* L.), papa China (*Colocasia esculenta* (L.) Schott), camote (*Ipomoea batatas* (L.) Lam.), oca (*Oxalis tuberosa* Molina), and mashua (*Tropaeolum tuberosum* Ruiz & Pavón) (Figure 1).

These roots have a high amount of water in a fresh state (>72.7%), which makes them highly perishable (Table 1). In nutritional terms, these roots contain considerable amounts of protein (3.11–5.74%) and fiber (0.9–3.41%) compared to potatoes (1–2% and 0.5%, respectively) [2]. They are an excellent source of carbohydrates (8.5–15.09%), which is why residents of the Andean areas widely consume them. The most traditional way to consume them is cooked or fried. Also, zanahoria blanca, achira, papa China, and camote contain a considerable amount of starch (42.82, 52.18, 59.98, and 40.05, respectively) (Table 1). This characteristic is of great technological interest since starch is the basis for making pasta, baked food, and extrusions.

Also, these roots contain high amounts of minerals (Ca, Cu, Fe, Mn, Mg. P, K, and Zn), vitamins (A, B, C, E), and fructo-oligosaccharides (FOS), which are considered prebiotics [3]. Some scientific studies have demonstrated high antioxidant activity due to polyphenols, anthocyanins, and flavonoids; consequently, they have curative effects against urinary disorders, asthma, arthritis, and diarrhea [4]. A significant benefit is the absence of gluten, which is crucial for those intolerant to these proteins [4].


foods-13-00051-t001_Table 1Table 1Mean values and standard deviations of chemical composition (% wb) of Andean roots and raw flours.

MoistureProteinFatAshCarbohydratesFiberStarch
**Ax**
Root ^a^76 (2)4.76 (0.86)2.2 (0.3)2.3 (0.3)15.1 (2.3)3 (1)14 (2)Flour ^f^6.2 (0.2)2.1 (0.2)0.69 (0.08)4.5 (0.2)77.3 (0.5)9.25 (0.02)42.8 (0.6)
**Ci**
Root ^b^73.04 (0.33)3.93 (0.04)3.40 (0.04)2.70 (0.02)14.15 (0.02)2.78 (0.03)13.63 (0.52)Flour ^f^5.9 (0.2)4.8 (0.2)0.63 (0.29)8.04 (0.04)69.09 (0.04)11.51 (0.05)52.18 (0.72)
**Ce**
Root ^b^72.7 (0.2)5.7 (0.2)2.2 (0.2)2.96 (0.26)13.2 (0.2)3.4 (0.2)12.2 (0.2)Flour ^f^6.22 (0.26)8.4 (0.2)0.73 (0.08)5.55 (0.02)64.2 (0.2)14.90 (0.02)59.98 (0.82)
**Ib**
Root ^c^74.2 (0.2)5.6 (0.2)1.1 (0.2)1.9 (0.2)13.7 (0.2)3.4 (0.2)12.7 (0.2)Flour ^f^6.20 (0.27)4.65 (0.24)0.37 (0.06)3.24 (0.49)74.35 (0.36)11.2 (0.2)40.05 (0.55)
**Ot**
Root ^d^79.22 (0.62)3.11 (0.02)1.71 (0.03)1.29 (0.03)13.01 (0.02)1.66 (0.08)9.6 (0.2)Flour ^f^16.4 (0.2)1.63 (0.07)1.06 (0.06)3.04 (0.24)72.5 (0.2)5.33 (0.05)28.12 (0.39)
**Tt**
Root ^e^82.8 (0.2)5.5 (0.2)1.7 (0.2)0.6 (0.2)8.5 (0.2)0.9 (0.2)7.1 (0.2)Flour ^f^18.87 (0.06)9.12 (0.13)0.59 (0.07)4.94 (0.05)56.9 (0.3)9.60 (0.05)22.2 (0.3)^a^ Matsuguma et al., 2009 [5]; ^b^ Tresina et al., 2020 [6]; ^c^ Mohanraj and Sivasankar 2014 [7]; ^d^ Jimenez, et al., 2015 [8]; ^e^ Apaza et al., 2020 [9]; ^f^ Salazar et al., 2021 [10] (dried at 60 °C for ~6 h).


However, although South America has crops rich in nutrients and with excellent technological characteristics, their qualities are not taken advantage of in the development of food. According to statistical data, although the cost of acquiring a healthy diet is similar between South America and the rest of the world (approximately USD 3.5 per person per day) (Figure 2a), only 18% of the South American population is able to afford this daily cost (Figure 2b). This statistic is alarming, especially when compared to the approximately 43% reported worldwide. This shows the need to develop more accessible (cheap) foods for this population.

In addition, an alarming increase in food insecurity is evident in the South American regions (approximately 3% more compared to 2020–2022) (Figure 3a). This is due to decreased government initiatives during the postpandemic economic crisis [12]. Furthermore, the prevalence of undernutrition is more concerning and reached 13.9% in the last period (2020–2021) (Figure 3b). This value is another important indicator, as it shows the lack of knowledge of the nutritional properties of the region’s native crops. Sadly, many South American farmers sell the roots, vegetables, and fruits they grow to buy processed foods, which are usually less nutritious (such as potatoes, rice, pasta, and bread). This shows the need to value the crops developed locally the through research and development of processed products.

Under the previously stated context, it is evident that South America presents severe problems in terms of food security and malnutrition. The main drawback is diet, which depends on wheat, corn, rice, and potatoes. It is contradictory that despite South America being a region with a diversity of vegetables and fruits, the excellent nutritional qualities of these crops are not valued. In this sense, one challenge facing the food industry today is the development of nutritious foods that are attractive in the market. Children and youth are among the most challenging sectors to tackle because healthy food is often associated with less tasty options [13]. Therefore, this study proposes the development of expanded snacks because they are an attractive food for these populations.

Extrusion is broadly used due to its versatility and low-cost food production. This technology involves a high-temperature (180–190 °C) short-time process (20–40 s), and it can transform a wide variety of raw materials into intermediate and finished products [14]. There are three types of extruded snack foods:First-generation snacks (1G) are minimally processed by roasting and frying. They are used for snacking (i.e., eating small amounts of food between meals) and include, for example, nuts, potato chips, and popped popcorn [15].Second-generation snacks (2G) are the most popular and commonly called “ready-to-eat” or “three-dimensional” snacks. These snacks are directly expanded through the extrusion exit [16]. Examples of snacks in this category include puffed corn, onion rings, and flavored loops.Third-generation snacks (3G), called “pellets” or “glassy half-products”, are not expanded directly through the extrusion exit and must be expanded before being consumed using additional processes, such as frying or microwave heating. These pellets are shelf-stable without refrigeration due to their low moisture (around 10% wb) as long as their packaging prevents moisture from increasing [17]. These snacks are made for export due to their high bulk density and stability.

Of the three types of extrudate snacks described above, the second-generation snacks (2G) were chosen in this study because they are more attractive to children, especially if there are plans to include these foods as part of a school breakfast or midmorning snack in future government policy [18].

The objectives of this work were (1) to manufacture second-generation (2G) snacks (product obtained from the immediate expansion of the mixture at the exit of the extruder die) with the addition of Andean tubers and tuberous root flour to increase the nutritional characteristics of this type of food and (2) to explore the influence of these flours on water content (x_w_), water activity (a_w_), the sectional expansion index (SEI), hygroscopicity, bulk density, porosity, the water absorption index (WAI), the water solubility index (WSI), the swelling index (SWE), texture, optical properties, and bioactive compounds. The study is expected to demonstrate the possibility of obtaining healthy snacks and to promote the appreciation of native Andean crops in the South American region and worldwide.

## 2. Materials and Methods

### 2.1. Raw Materials

Zanahoria blanca (*Arracacia xanthorrhiza* Bancr.), achira (*Canna indica* L.), papa China (*Colocasia esculenta* (L.) Schott), two varieties of sweet potato (*Ipomoea batatas* (L.) Lam.) (purple and yellow), three varieties of oca (*Oxalis tuberosa* Molina) (white, yellow, and red), and mashua (*Tropaeolum tuberosum* Ruiz & Pavón) were purchased from a local market (Ambato, Ecuador). The Andean crops were selected based on a previous critical analysis (SWOT) of the tubers most undervalued in the locality since long-term research seeks to enhance their use as an ingredient in food products. Maicerías Españolas S.L. (València, Spain) supplied the corn grits.

### 2.2. Flour Manufacturing

Ten roots of each type with no physical defects were selected. The roots were washed, peeled, and cut into slices (2 ± 0.1 mm). The drying process was conducted in a convective dehydrator (Gander mtn, CD 160, Saint Paul, MN, USA) at 60 °C for 24 h. The dehydrated products were ground in an electric mill (Hamilton Beach, model: 80393, Picton, ON, Canada) at three intervals of 10 s. The flour was preserved in separate airtight aluminized bags at 25 °C.

### 2.3. Production of 2G Snacks

Ten different samples were prepared (one control and nine mixtures of corn + root flour). For the control, 100% corn grits were used. For the remaining nine samples, the corn grits were mixed with each root flour in a proportion of 80% corn grits to 20% of the samples’ flour. The following samples were obtained: C, control (corn grits); Ax, zanahoria blanca flour; Ci, achira flour; Ce, papa China flour; IbP, purple sweet potato flour; IbY, yellow sweet potato flour; OtW, oca white variety flour; OtY, oca yellow variety flour; OtR, oca red variety flour; and Tt, mashua flour.

A laboratory extruder (single-screw KE 19/25; length-diameter ratio of 25:1; nozzle diameter 3 mm; Brabender, Duisburg, Germany) was used to obtain 2G snacks. The process was conducted at a compression ratio of 3:1, and the samples were fed at a constant speed of 150 rpm. The rotation speed was 120 rpm, and temperatures of the barrel were as follows: Section 1, 25 °C; section 2, 70 °C; section 3, 170 °C; and section 4, 175 °C. The equipment was monitored through the Extruder Winext software, version 4.4.3 (Brabender, Duisburg, Germany). The 2G snacks were preserved in airtight aluminized bags at 25 °C.

### 2.4. Characterization of 2G Snacks

#### 2.4.1. Water Content (x_w_) and Water Activity (a_w_)

A vacuum oven (Vaciotem, J.P. Selecta, Barcelona, Spain) was used (103 °C for 48 h) to obtain the x_w_. A hygrometer (AquaLab PRE, Decagon Devices, Inc., Pullman, WA, USA) was used to determine the a_w_. An average of 3 measurements were made in each characteristic.

#### 2.4.2. Sectional Expansion Index (SEI)

A digital caliper was used to measure the width and length of the 2G snacks. The SEI was determined using the methodology proposed by Patil et al. [19]. An average of 10 measurements were made.

#### 2.4.3. Hygroscopicity (Hy)

Hygroscopicity is the capacity of a food to absorb water. At an industrial level, food low in hygroscopicity is desirable since this characteristic makes it less perishable [20]. The method proposed by Cai and Corke [21] was used to determine the Hy. The samples were positioned in glass Petri dishes in a desiccator with saturated Na_2_SO_4_ solution (at 25 °C and 81% relative humidity). After seven days, each sample was weighed with a precision balance (±0.001 g) (Mettler Toledo, Greifensee, Switzerland). The results are expressed as grams of water gained per 100 g _dry solids_. An average of 3 measurements were made.

#### 2.4.4. Bulk Density (ρ_b_) and Porosity (ε)

Bulk density (ρ_b_) is the ratio between the total mass and the total volume, including the air contained in that volume [22]. The method proposed by Gujska and Khan [23] was used to determine ρ_b_. It was calculated by dividing the mass of the product (weighed with a precision balance ((±0.001 g) Mettler Toledo, Greifensee, Switzerland) by the volume (measured with an electronic Vernier caliper; Comecta S.A., Barcelona, Spain) and expressed as g/m^3^. The porosity (ε) was calculated from density (ρ) (determined with a helium pycnometer; AccPyc 1330, Micromeritics, Norcross, GA, USA) and bulk density (ρ_b_), according to García-Segovia et al. [24]. An average of 10 measurements were made in each characteristic.

#### 2.4.5. Water Absorption Index (WAI), Water Solubility Index (WSI), and Swelling Index (SWE)

The methods proposed by Singh and Smith [25] and Uribe-Wandurraga et al. [26] were used to measure the WAI and WSI, respectively. The snacks were milled (average particle size of 180 to 250 µm), and 2.5 g of milled sample was dispersed in 25 g of distilled water. The mixture was stirred for 30 min using a magnetic stirrer and placed in centrifuge tubes (50 mL) until reaching a weight of 32.5 g. The tubes were centrifuged at 3000 rpm for 10 min. Finally, the supernatant was decanted to determine its dissolved solids content, and the sediment was weighed. The results of WAI and WSI were calculated with Equations (1) and (2), respectively.
(1)$$WAI=\frac{weight\;of\;sediment}{weight\;of\;dry\;solids}$$
(2)$$WSI\;(\%)=\left(\frac{weight\;of\;dissolved\;solids\;in\;supernatant}{weight\;of\;dry\;solids}\right)\times 100$$

The method detailed by Robertson [27] was used to measure SWE. One gram of milled sample (particle size: 180–250 µm) was placed in graduated cylinders, and 10 mL of distilled water was added. The mixture was left to rest for 18 h, and the volume reached was recorded. The result is expressed as the milliliters of swollen sample per gram of dry initial sample. An average of 3 measurements were made in each characteristic.

#### 2.4.6. Texture Properties

Texture properties were measured using a TA-XT2 Texture Analyzer (Stable Micro Systems Ltd., Godalming, UK). Puncture tests (speed 0.6 mm/s) were performed on cylinders of 2 mm in diameter. The area under the force–time curve plot was determined using Texture Exponent software (version 6.1.12.0). The properties measured were the average specific force of structural ruptures (F_s_), the average puncturing force (F_p_), the spatial frequency of structural ruptures (N_sr_), the number of peaks (N_0_), and the crispness work (W_c_) [28,29]. An average of 3 measurements were made in each characteristic.

#### 2.4.7. Optical Properties

Optical properties were measured with a Minolta spectrophotometer (CM-3600d, Tokyo, Japan) using the standard light source D65 and a standard observer 10. The CIE*L*a*b* color coordinates (spectra 400 to 700 nm) were considered, and the parameters determined were luminosity (L*), chroma (C*), hue (h*), and the total color difference (ΔE) between the mixture before extrusion and the 2G snack. An average of 3 measurements were made in each characteristic.

#### 2.4.8. Bioactive compounds

Total carotenoids (TC)

The method proposed by Olives Barba et al. [30] was used to determine total carotenoids (TC). The AOAC spectrophotometric method [31] was used to quantify TC and is expressed as the milligrams of β-carotene per 100 g of sample.

Lycopene (LP)

This property was determined (at 503 nm) using the TC extract and is expressed as milligrams per 100 g of the sample [32].

Total phenols (TP)

Igual et al. [33] described the method for measuring total phenols (TP). A UV-3100 PC (VWR, Radnor, Philadelphia, PA, USA) was used to measure the absorbance of the sample at 765 nm, and the TP is expressed as milligrams per gallic acid/100 g of dry solid sample.

Antioxidant capacity (AC)

The 2,2-diphenylpicrylhydrazyl (DPPH) method described by Igual et al. [33] was used to determine antioxidant capacity (AC). The absorbance was measured at a wavelength of 515 nm, and the results were expressed as milligram Trolox equivalents (TE) per 100 g of dry solid sample. An average of 3 measurements were made in each characteristic.

### 2.5. Statistical Analysis

Statgraphics Centurion XVII software (version 17.2.04; Statgraphics Technologies, Inc., The Plains, VA, USA) was used for statistical analysis. Analysis of variance (ANOVA) and Tukey’s multirange test were applied, with a confidence level of 95% (*p* < 0.05), to evaluate differences among snacks. A Pearson correlation analysis was performed between the bioactive compounds analyzed and the antioxidant capacity determined, with a significance level of 95%.

## 3. Results and Discussion

### 3.1. Characterization of 2G Snacks

Table 2 shows the results found in a_w_, the sectional expansion index (SEI), hygroscopicity (Hy), bulk density (ρ_b_), porosity (ε), the WAI, the WSI, and the swelling index (SWE) for all 2G snacks and control sample.

#### 3.1.1. Water Content (x_w_) and Water Activity (a_w_)

Extrusion is a typical food industry process used to produce cooked food by exposing the materials to high-pressure (1.0 to 5.1 MPa), high-shear, and high-temperature (>150 °C) environments [34]. Due to the temperature reached in the process (170 and 175 °C), the extrusion considerably reduced the x_w_ in all the samples (Figure 4). According to some studies, the moisture content is reduced by approximately 60% during extrusion. For example, in an extruded snack made with maize and lupine (*Lupinus albus* L.) flour (ratio 80:20, processed at 150 °C), a reduction from 16% moisture in the mixture to 6.96 ± 0.61% moisture in the snack was observed [35]. Also, a study on maize grits showed a reduction from 10.94% moisture in the mixture to 5.05% moisture in the snack [36].

The samples that showed the most significant reduction were OtW, OtY, and OtR (*p* < 0.05). This effect was also demonstrated in third-generation (3G–pellets that are ready to be consumed after a secondary expansion extrusion), microwave-expanded snacks [37] manufactured from Andean tubers and tuberous root flours [38]. However, the samples that showed less variation were C and Tt (Figure 4). This means that x_w_ variations are influenced by root composition. The corn grits and Tt root have a higher amount of protein (8.61 ± 0.01% [39] and 9.12 ± 0.13% [10], respectively) compared to Ot root (1.63 ± 0.07%) (Table 1) [10]. This could indicate that protein plays an essential role during extrusion.

The a_w_ values of the 2G snacks were >0.5 (Table 2). Statistically significant differences were observed among snacks made with roots of the same species (*p* < 0.05). For example, OtR (0.593 ± 0.003) had lower values compared to OtW (0.645 ± 0.003) and OtY (0.674 ± 0.003). A similar effect was observed between IbY (0.585 ± 0.003) and IbP (0.606 ± 0.003). This parameter represents the “water availability” and indicates how perishable and highly hygroscopic foods are. These values are essential for selecting the packaging and storage for food industrialization. Some studies have shown that snacks lose crispness when a_w_ exceeds 0.5 and lose brittleness at 0.8 [40,41]. The high a_w_ values emphasize the need to pack these snacks in containers that prevent moisture, such as biaxially oriented films and films coated with other polymers or aluminum to improve the barrier properties [42].

Also, a_w_ depends on the composition of the food solute. Foods have high a_w_ values when their composition contains a large amount of high-molecular-weight solutes (proteins, cellulose, starch) [43]. In this case, the water molecules interact with the hydrophilic sites of these solute structures. This also means that polymeric solutes produce stable products regarding a_w_. According to the significant differences observed (*p* < 0.05), it can be inferred that this occurred in the C, Ci, IbY, OtR, and Tt samples. On the other hand, carbohydrates with low molecular weight (glucose, fructose, etc.) interact and dissolve in water through hydrogen bonds, resulting in fewer free water molecules being available [44]. Likewise, it can be inferred that this occurred in the Ax, Ce, IbP, OtW, and OtY samples. Finally, it could be summarized that the snacks obtained from these roots have a different solute composition, which affects different values of a_w_.

#### 3.1.2. Sectional Expansion Index (SEI)

The 2G snacks examined in this investigation had an SEI value between 10.4 and 12.1 (Table 2). These values can be considered high compared to snacks made with corn flour with the addition of amaranth, quinoa, and kañiwa flour (ratio of 80% corn flour and 20% crop flour), which had SEI values of 7.6, 6.1, and 5.1, respectively. These values may be attributable to the plasticizing effect of monosaccharides, oligosaccharides, amines, and water in these crops [45].

Tubers Ax and OtY provides snacks a higher SEI value; however, samples Ci and IbP had the lowest SEI (Table 2). Some studies have shown that higher proportions of fiber in the mixtures cause lower extrudate expansion [46,47]. The insoluble fiber components, such as cellulose and hemicellulose, have a high content of hydroxyl groups that can link with water. Consequently, there is insufficient water to complete starch gelatinization [48], and the lower the gelatinization degree is, the less expansion is observed in the final product [49]. This could explain why the Ot samples showed lower SEI values compared to the other samples. The Ot roots (*Oxalis tuberosa* Molina) had lower fiber values (5.33 ± 0.05%) [10] compared to root Ci (11.51 ± 0.05%) and IbP (11.19 ± 0.11%) (Table 1) [10].

#### 3.1.3. Hygroscopicity (Hy)

In this study, all snacks comprised sugars, pectin, and cellulosic material (due to the nature of the corn grits and the Andean roots used—Table 1), which have an affinity for water vapor. The Ax, IbP, and IbY products had higher hygroscopicity values (Table 2). These results could indicate that these snacks have substantial carbohydrates with low molecular weight (glucose, fructose, etc.), which causes increased hydration [50]. Research shows that the Ax root has 34.94% sugar while the IbP root has 34.3% [10]. The snacks with the lowest tendency to gain water are C and Ce. As previously discussed in the a_w_ results, the proportional trend was observed between the hydration capacity and the amount of sugars in the raw materials. Furthermore, studies have shown that corn grits (C: control) have 1.64% sugar [39], whereas Ce roots have 4.28% sugar [10]. This parameter is also relevant because a notable loss of crispness is usually observed when the moisture content of low-moisture products increases due to their hygroscopic capacity. Water absorption influences the mobility and flexibility of molecules, increasing chemical degradation reactions and altering the texture of the snack (which reduces hardness) [51]. Therefore, the snacks developed in this research must be packaged in containers that prevent the permeability of water vapor from the environment to preserve their characteristic textural properties.

#### 3.1.4. Bulk Density (ρ_b_) and Porosity (ε)

These properties are usually used to evaluate the extruded expansion in all directions. The 2G snacks examined in this investigation had bulk density values between 0.084 and 0.16 g/cm^3^ (Table 2). These values are similar to those obtained in snacks prepared with brown rice grits (0.06–0.14 g/cm^3^) [52], rice and corn flour with spirulina addition (0.071–0.185 g/cm^3^) [53], and corn flour with microalgae addition (0.079–0.105 g/cm^3^) [26].

The 2G snacks with the highest ρ_b_ were IbP and Tt. In contrast, the snacks with the lowest values were C, Ax, Ci, and OtR. Some studies reported an inversely proportional relationship between the mixtures’ fat content and the snacks’ bulk density [24]. This would explain the low values obtained in the IbP and Tt snacks, as these roots had fat contents of 0.37 ± 0.06% and 0.59 ± 0.07% (Table 1) [10], respectively, which were lower than those of the corn grits (2.62 ± 0.02%) [39] and Ot (1.06 ± 0.06%) [10]. The density did not show an evident relationship with the values determined for porosity (%).

The porosity values ranged from 84.86% to 92.42% (Table 2). The addition of root flour reduced the porosity. The samples observed with the least porosity were Ce, Tt, and IbP. According to previous studies, there is an inversely proportional relationship between the protein content of the mixtures and the snack porosity [54,55]. During extrusion, proteins change, exposing the hydrophobic amino acids initially enclosed. These hydrophobic amino acids compete for water with starch and produce aggregation through protein–protein and protein–water links. Finally, water evaporation occurs, which forms an inflated structure sustained by protein crosslinking [56,57]. These protein alterations influence the change in viscosity, gelation, solubility, and textural properties, which would explain the low porosity values obtained in the Ce, Tt, and IbP snacks, whose raw root flours demonstrated protein contents of 8.37 ± 0.14%, 9.12 ± 0.13%, and 4.65 ± 0.24% [10], respectively [10]. Other factors, such as raw protein type, pretreatment, and extrusion conditions, can also play an essential role [58].

#### 3.1.5. Water Absorption Index (WAI), Water Solubility Index (WSI), and Swelling Index (SWE)

The WAI showed significant differences between the samples (*p* < 0.05) (Table 2). The IbP, OtY, and Tt snacks had the highest values. This property measures the snack’s water absorption capacity, which affects its melt, break-up, and distend power after extrusion [41]. The more aggressive the process is, the greater the breakage of inter- and intramolecular hydrogen bonds, resulting in more exposed hydroxyl groups [59]. Therefore, it can be inferred that the IbP, OtY, and Tt samples underwent greater starch degradation during the process. Finally, a direct proportional relationship was observed between bulk density, porosity, and WAI in the IbP and Tt samples. A similar direct proportional relationship was observed in the puffed rice snacks [60].

The WSI reflects the amount of soluble compounds released from the starch granules during extrusion, generating higher solubility values [61]. It can be inferred that the starch of samples C, Ax, Ci, Ce, and OtR underwent a more remarkable disintegration during processing according to the high WSI values observed (Table 2). In addition to the changes that starch undergoes during extrusion, this process reduces the molecular weight of polysaccharides (such as pectin and hemicellulose), producing more soluble components [62]. Some studies have demonstrated that even when the fiber content does not change considerably, the extrusion solubilizes some fiber components, and increases in soluble dietary fiber can be observed [63,64]. The lowest values were observed in the IbP and Tt snacks. Also, the extrusion process increases the exposure of hydrophobic groups due to protein denaturation, which reduces the extruded solubility in aqueous systems. For example, it releases the amino acids alanine, valine, leucine, isoleucine, tyrosine, and phenylalanine, which are present in IbP [65] and Tt [66]. This effect was studied by Chiang [67], who demonstrated the importance of the protein–protein interaction during the process. During extrusion, soy isolate protein develop new chemical bonds or cross-linkages (hydrogen bonds and hydrophobic interactions). These bonds and links participate in the texture and fibrous structure, which are characteristic of the protein network and reduce the solubility of extrudates [68].

The SWE showed significant differences among the samples (*p* < 0.05) (Table 2). The highest SWE values were found the IbP and Tt samples. In this parameter, starch gelatinization plays an essential role. The raw starch is transformed into cooked and digestible material in this conversion. Furthermore, the starch molecules can be divided through a process called dextrinization. All these internal changes generate greater swelling power in the snacks. Likewise, a direct proportional relationship was observed between the WAI and SWE in all samples except for OtY. A similar effect was observed in a study on optimizing the extrusion process of soybean hull [69]; the authors explain this is because both reflect the hydration properties of soluble components such as sugars and fibers. Therefore, a deeper predictive model needs to be conducted between the composition of the roots and its interrelation with the operating conditions to optimize the process.

#### 3.1.6. Texture Properties

F_s_ expresses the strength spent to break each bubble of air in the snack structure. The addition of root flour reduced F_s_, except for IbP, for which the value was significantly higher (*p* < 0.05) (Table 3).

The average puncturing force (F_p_) is the most direct sensory index associated with hardness. Low values indicate less force required to initially bite through the snack. In this study, the softer snacks during the first bite were IbY, OtY, OtR, and Ax.

N_sr_ describes the number of peaks along the distance traveled by the sensor. This parameter reflects phenomena during extrusion, such as nucleation, extrudate swelling, bubble growth, and bubble collapse [70]. In this study, two important groups were observed. Snacks made with C and Ci showed the significantly lowest values (*p* < 0.05). This can be observed in Figure 5, where the sections show solid areas in both samples and a smaller number of collapsed bubbles. N_0_ indicates the number of bubble air pockets present in the structure. Lower values were observed in the C and IbP samples. The remaining samples had high values and no significant differences (*p* > 0.05). However, this property should be correlated with F_p_ values because numerous cells with low F_p_ indicate good crispness in this type of food [71]. Samples Ax, IbY, OtW, OtY, and OtR had better crunchiness.

The crispness work (W_c_) values ranged between 0.16 and 0.82 N.mm. The addition of root flour reduced W_c_, except for the IbP sample, in which the value was significantly higher (*p* < 0.05) (Table 3). This textural parameter is associated with the initial sound produced by the snack during the first bite [72]. It is essential because snacks are desired for their crispness and the pleasant sound produced when bitten.

The parameters W_c_, F_s_, and F_p_ showed higher values in IbP and lower values in IbY. As discussed in Section 3.1.1, statistically significant differences were observed between snacks made with roots of the same species but different varieties (*p* < 0.05). The evidence shows that the textural properties are related to protein denaturation. Furthermore, some studies have shown that a reduced amount of lipid (<5%) facilitates steady extrusion and improves texture.

The presence of crude fiber increases the water absorption of food, giving it an excellent final texture [73]. This study highlights the necessity for a more thorough analysis and microscopy of the composition of roots used. Furthermore, a more detailed evaluation of the starch granules and fiber is essential to define the most influential parameters in this textural property.

#### 3.1.7. Optical Properties

The extrusion process significantly reduced the lightness values (L*) in all the samples analyzed (*p* > 0.05) (Table 4). The samples with higher values of L* were C, Ce, and OtW. This could be related to these roots having whitish pulps (Figure 6). In contrast, lower values were obtained in the IbP, IbY, and Tt samples. Apart from the more saturated colors presented by the pulps of these roots, this trend could indicate that the mixtures in these treatments had lower values of starch content and feed moisture, which usually develop low L* values and darker products [74].

The extrusion process increased the chroma values (C*) in the Ce, IbP, Ax, and OtR samples. The extrusion process can explain this, as the high-temperature cooking releases free sugars (glucose and fructose) from starch hydrolysis. Furthermore, some soluble fiber fragments can be found after this process. Both compounds are precursors for the Maillard reaction with proteins, which causes more saturated colors in snacks due to the formation of melanoidins (brown polymers). However, OtW, OtY, and C exhibited less saturated colors. These samples also had low WAI and SWE values (Table 3). Therefore, it can be inferred that these samples released less soluble compounds, such as sugars and fibers, during the extrusion process.

The hue (*h) values showed significant differences between the process and the samples (*p* < 0.05) (Table 4). Higher values were observed in the mixtures compared to the snacks obtained. This parameter is related to the type of color. Snacks are mostly grouped in the CIE*L*a*b* space based on similar h* values. Samples C and Ce had higher values (82.1 ± 0.06 and 75.67 ± 0.07, respectively), whereas the OtY and Tt samples had the lowest values (71.77 ± 0.06 and 70.8 ± 0.2, respectively). This is evident visually in Figure 6, where the OtY and Tt samples have slightly different colors than do the other samples.

The samples that showed the most remarkable color variation (between the mixture and the 2G snacks) were OtW, OtY, OtR, and Ax (*p* < 0.05). In contrast, the samples with the least variation were C, Ci, and Tt. From the images in Figure 6, it can be inferred that when the mixtures had less saturated colors, and they generate more notable color changes in the snack. In contrast, darker and more saturated mixtures (C, Ci, and Tt) caused fewer overall color variations (ΔE). Even if the snack had different colors, all the samples developed satisfactory colored products under the processing conditions of this study.

The properties previously discussed demonstrate the excellent technological characteristics of 2G snacks. They have low water content and their a_w_ makes them a nonperishable food. Also, Andean root flours achieved higher SEI and ρ_b_ than did the control snacks (100% corn grits). They had a higher WAI value and a lower WSI value than did the control snacks. Considering processed snacks can be consumed for breakfast, these characteristics are interesting when analyzing the conservation of its crunchy features when immersed in aqueous bases (milk or yogurt) [75]. Furthermore, adding Andean root flour improved the snack’s textural characteristics in terms of crunchiness and crunchiness. It generated more brown saturated colors, which can be attractive to the consumer since it can be related to more natural and integral products [76].

This shows the viability of using Andean flour and roots in preparing snacks that consumers widely accept. Furthermore, as they are undervalued crops, the cost of marketing in the Andean agricultural sector is lower than it is for potatoes, rice, and corn. For example, 100 kg of roots such as Ax, Ot, and Tt usually cost USD 4, while 100 kg of potatoes generally reach values of up to USD 30 [77], rice around USD 42 [78], and corn USD 18 [79].

The excellent technological characteristics of these 2G snacks and the considerably lower price of the Andean roots (raw material) constitute two great strengths that the food industry must consider when deciding to develop this type of food.

#### 3.1.8. Bioactive Compounds

The results of bioactive compounds characterized as total carotenoids (TC), total phenols (TP), and antioxidant capacity (AC) are shown in Table 5. Undervalued or unknown Andean crops represent a challenge because the results are not easily comparable to those in the literature, as there is insufficient information on how to use them for food production. However, there is information on crops’ nutritional, antioxidant, or technological properties, such as raw material or flour. In this sense, the results obtained in this study could help to understand the behavior in extrusion operations.

The TC values showed a significant variation (*p* < 0.05) between 1.75 and 4.74 mg_βcarotene_/100 g of a sample. Most snacks showed higher TC values than did the control snacks. The exception was found in the Ax, Ce, and IbP snacks. In this study, OtY had the highest carotenoid value (4.74 mg_βcarotene_/100 g of sample) (*p* < 0.05); this value was also highest in the Tt sample, which corresponds to mashua root, reported as a crop with an essential carotenoids content [9,80]. The TC value in OtY in this study was 1.63 mg_βcarotene_/100 g, which is more that of a 3G yellow oca obtained in a previous study [38]. Furthermore, the values are greater than those reported by Campos [81], who studied 14 oca genotypes (0.2–0.25 mg_βcarotene_/100 g of sample). The lowest value was detected in Ax and C probably due to the reduced pigment content in contrast with the other crops. However, when comparing TC values between 3G snacks and 2G snacks, some observed that 2G snacks with purple camote and white oca had 0.42 and 0.22 mg_βcarotene_/100 g less than did the 3G snacks, respectively. However, the TC content was higher in 3G red oca and mashua snacks. These results are probably attributable to the nature of the components; furthermore, the different thermic treatments could influence the results.

The total phenol results showed that Ci had the highest value (130.6 mg_GAE_/100 g); it is essential to note that all extruded Andean crops had values above the control (*p* < 0.05). The results are comparable with those obtained by Praseptiangga [82], who reported a value of 125 mg_GAE_/100 g in boiled *Canna indica*. The total phenol could be attributed to phenolic acids, flavonoids, tannins, and the hydrolysis of different compounds, such as proteins, which release phenolic compounds and make them more available. Furthermore, it could be attributed to the degradation of anthocyanins that help form various polyphenolic compounds because the Folin–Ciocalteu reagent could react with proteins and sugars that are part of the composition of these tubers [10,83]. In this study, the extruded 2G snacks from Andean crops had values around 112–130 mg_GAE_/100 g. The Ce sample had the lowest value compared with the other Andean tubers; however, it is essential to note that these values (112 mg_GAE_/100 g) were the highest compared to those reported by Salazar [10] in taro flour. The phenol content of IbP, IbY, Ax, and Tt was similar to those reported by Catunta [84] in fresh mashua samples (128–146 mg_GAE_/100 g). However, in the OtW, OtY, and OtR snacks, the phenolic content values were similar to those reported by Campos [81] in fresh oca genotypes (71–132 mg_GAE_/100 g). Phenolic compounds, naturally occurring antioxidants, are abundant in various plant-based foods and beverages and play an essential role in nutritional and healthcare contexts [85,86].

The AC results showed that the highest value corresponds to the OtY snack; however, there was a slight difference with the Tt snack (*p* < 0.05). This is important because mashua eventually showed the best antioxidant activity of the Andean crops studied here and reported in the literature [10,38,87]. In contrast, Ce snacks had the lowest AC (*p* < 0.05) between Andean crops; Ce was higher than C. The AC is attributable to the phenolic compounds, flavonoids, carotenes, and vitamin C, which are reported to be part of these roots [88]. The evaluation of Pearson correlations between TC and TP with AC showed a significant difference (*p* < 0.05), and correlational statistical analyses showed the influence of TC and TP on the AC of the samples. TP was essential in AC (0.75, *p* < 0.05).

## 4. Advantages and Limits

The advantages of this study are that it aims to apply a different extrusion method to obtain snack-type products based on nonconventional raw materials (flour from Andean crops). Even though this technology has previously been tested for producing 3G snacks, this study shows the possibility of obtaining completely expanded snacks directly on the equipment. The population’s desire for foods based on new raw materials that are ready to eat generated certain expectations before carrying out this research. The authors examined the developed snacks to assess the possibility of producing technologically stable appetizers with better nutritional value than conventional products found on the market. Although this work shows progress in this type of product and in the potential to offer new products based on these tubers and tuberous roots in disuse, it is necessary in the future to cover the issue of sensory and consumer perception.

## 5. Conclusions

Extrusion is becoming essential in modern food processes because some previously published studies show its ability to cook, minimize nutrient loss, and texturize products cheaply. This study showed that adding root flour increased the x_w_, a_w_, SEI, Hy, ρ_b_, and the WAI, but decreased the ε. However, all 2G snacks manufactured with Andean root flour showed better characteristics in texture, optical properties, and bioactive compounds. From a textural point of view, the IbY, OtY, OtR, and Ax samples established better characteristics in terms of crispness and crunchiness. The Ce, IbP, Ax, and OtR samples showed intense and saturated colors due to this process. During extrusion, the free sugars produced by starch hydrolysis and proteins were precursors for the Maillard reaction, which causes saturated colors in snacks. From the point of view of bioactive components, the snacks made with the roots of Ci, IbY, Ot, OtY, OtR, and Tt stand out. It can be inferred that the composition of raw roots plays a crucial role, as previous studies have shown that compounds, such as starch, fiber, and protein, change through the extrusion process, exposing the hydrophobic points and producing aggregation through protein–protein and protein–water links, which affects the extrinsic properties of the snacks.

## Figures and Tables

**Figure 1 foods-13-00051-f001:**
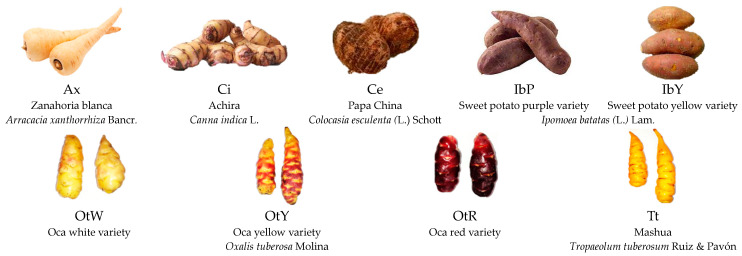
Andean tubers and tuberous roots.

**Figure 2 foods-13-00051-f002:**
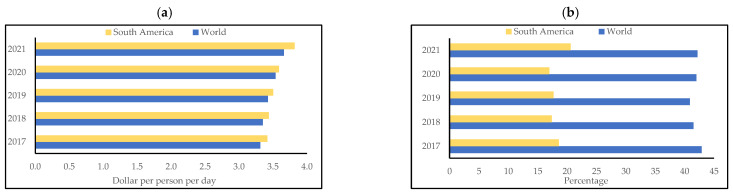
Comparison of (**a**) cost of a healthy diet (USD 1 per person per day) and (**b**) the percent of the population unable to afford a healthy diet worldwide and in South America. Source: FAOSTAT statistics database [11].

**Figure 3 foods-13-00051-f003:**
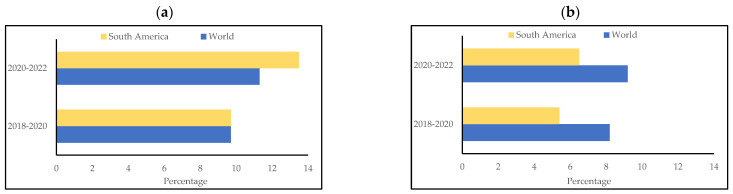
Comparison of (**a**) the prevalence of severe food insecurity in the total population (%) (3-year average) and (**b**) the prevalence of undernourishment (%) (3-year average) worldwide and in South America. Source: FAOSTAT statistics database [11].

**Figure 4 foods-13-00051-f004:**
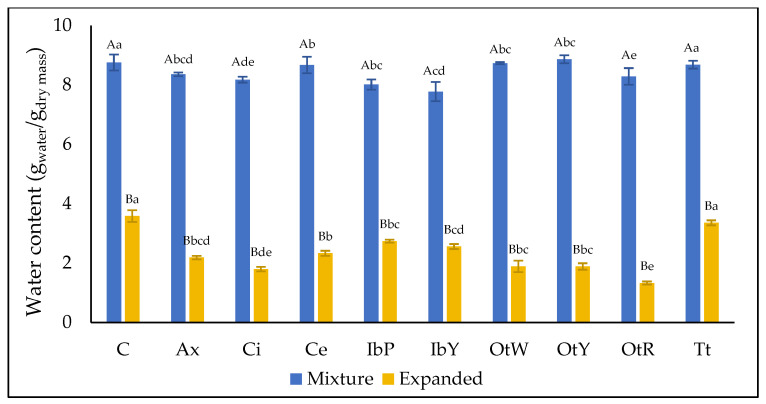
Mean values and standard deviations of the water content of the mixture and expanded snacks. Different capital letters represent significant differences (*p* < 0.05) by process, and lowercase letters represent significant differences (*p* < 0.05) by samples.

**Figure 5 foods-13-00051-f005:**
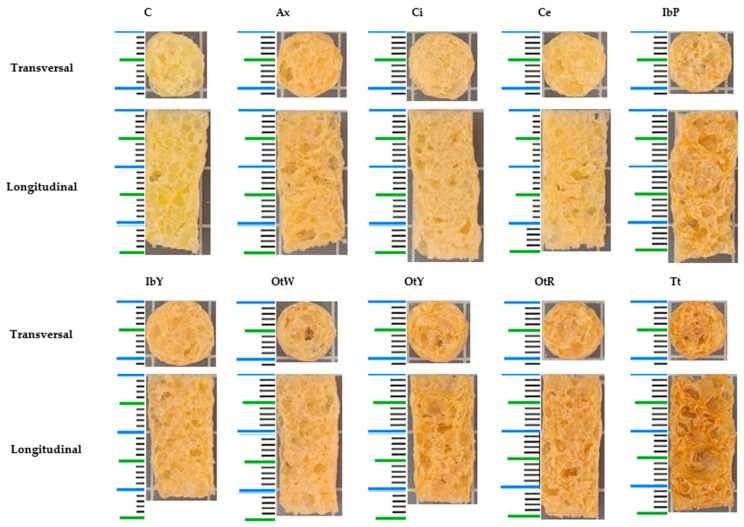
Transverse and longitudinal section of 2G snacks. Samples: C—control; Ax—zanahoria blanca; Ci—achira; Ce—papa China; IbP—purple sweet potato; IbY—yellow sweet potato; OtW—oca white variety; OtY—oca yellow variety; OtR—oca red variety; Tt—mashua.

**Figure 6 foods-13-00051-f006:**
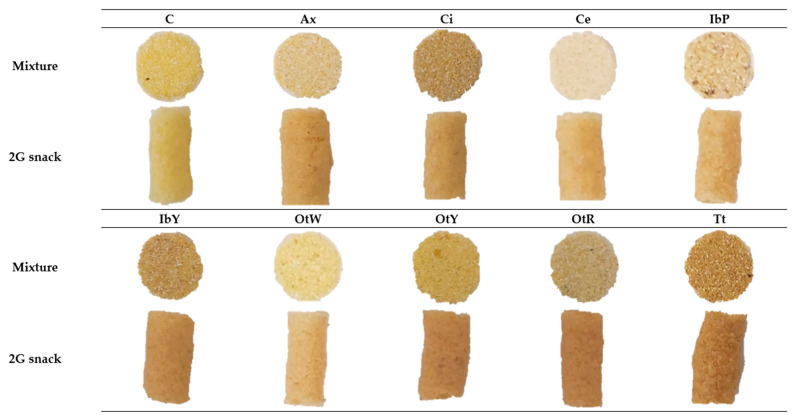
The appearance of mixtures and 2G snacks. Samples: C—Control; Ax—zanahoria blanca; Ci—achira; Ce—papa China; IbP—purple sweet potato; IbY—yellow sweet potato; OtW—oca white variety; OtY—oca yellow variety; OtR—oca red variety; Tt—mashua.

**Table 2 foods-13-00051-t002:** Mean values and standard deviations of the 2G snacks for measure parameters.

Sample	a_w_	SEI	Hy	ρ_b_	ε	WAI	WSI	SWE
			(g_w_/100 g _dry solid_)	(g/cm^3^)	(%)		(%)	(mL _swollen_/g _dry solid_)
C	0.567 (0.003) ^b^	11.1 (0.8) ^bcd^	11.5 (0.4) ^d^	0.085 (0.005) ^c^	92.4 (0.2) ^a^	4.73 (0.06) ^c^	11.57 (0.14) ^a^	4.03 (0.12) ^bcd^
Ax	0.665 (0.003) ^a^	12.1 (0.6) ^a^	17.2 (0.6) ^a^	0.084 (0.005) ^c^	91.9 (0.2) ^ab^	4.65 (0.07) ^c^	12.1 (0.3) ^a^	4.1 (0.3) ^bcd^
Ci	0.525 (0.003) ^b^	10.4 (0.8) ^d^	14.563 (0.107) ^bc^	0.08 (0.007) ^c^	90.3 (0.6) ^bc^	4.72 (0.05) ^c^	11.6 (0.2) ^a^	4.32 (0.12) ^bcd^
Ce	0.631 (0.003) ^a^	11.6 (0.9) ^ab^	13.6 (0.7) ^c^	0.11 (0.002) ^b^	89.3 (0.8) ^c^	4.72 (0.04) ^c^	11.960 (0.002) ^a^	3.67 (0.08) ^d^
IbP	0.606 (0.003) ^a^	10.6 (0.7) ^cd^	17.0 (0.4) ^a^	0.156 (0.003) ^a^	84.9 (0.7) ^e^	5.27 (0.02) ^a^	5.68 (0.17) ^d^	6.1 (0.4) ^a^
IbY	0.585 (0.003) ^b^	11.14 (0.8) ^bc^	17.0 (0.6) ^a^	0.095 (0.006) ^bc^	91.5 (0.5) ^ab^	5.03 (0.07) ^b^	9.59 (0.12) ^b^	4.3 (0.2) ^bcd^
OtW	0.645 (0.003) ^a^	11.1 (0.7) ^bcd^	15.0 (0.3) ^b^	0.098 (0.004) ^bc^	90.82 (0.05) ^abc^	5.00 (0.07) ^b^	9.6 (0.3) ^b^	4.59 (0.19) ^bc^
OtY	0.674 (0.003) ^a^	11.6 (0.8) ^ab^	15.6 (0.6) ^b^	0.097 (0.002) ^bc^	90.59 (0.13) ^bc^	5.26 (0.03) ^a^	8.0 (0.6) ^c^	3.94 (0.15) ^cd^
OtR	0.593 (0.003) ^b^	11.1 (0.7) ^bcd^	15.4 (0.3) ^b^	0.09 (0.006) ^c^	91.4 (0.3) ^ab^	4.55 (0.04) ^c^	12.4 (0.5) ^a^	3.98 (0.12) ^cd^
Tt	0.546 (0.003) ^b^	11.2 (0.4) ^bc^	15.691 (0.159) ^b^	0.16 (0.003) ^a^	87.54 (0.05) ^d^	5.29 (0.02) ^a^	5.9 (0.5) ^d^	4.76 (0.03) ^b^

C—control; Ax—zanahoria blanca; Ci—achira; Ce—papa China; IbP—purple sweet potato; IbY—yellow sweet potato; OtW—oca white variety; OtY—oca yellow variety; OtR—oca red variety; Tt—mashua; a_w_—water activity; SEI—sectional expansion index; Hy—hygroscopicity; ρ_b_—bulk density; ε—porosity; WAI—water absorption index; WSI—water solubility index; SWE—swelling index (SWE). Different letters in the columns represent significant differences (*p* < 0.05) by samples.

**Table 3 foods-13-00051-t003:** The texture parameters (mean values and standard deviations) of expanded snacks.

Sample	F_s_ (N)	F_p_ (N)	N_sr_ (mm^−1^)	N_0_	W_c_ (N.mm)
C	3.1 (0.4) ^b^	2.2 (0.3) ^b^	6.1 (0.5) ^d^	61 (5) ^c^	0.50 (0.03) ^b^
Ax	1.6 (0.2) ^f^	1.2 (0.2) ^c^	6.7 (0.4) ^bcd^	71 (5) ^ab^	0.23 (0.03) ^e^
Ci	2.5 (0.12) ^cd^	2.01 (0.08) ^b^	6.2 (0.5) ^d^	70 (5) ^ab^	0.40 (0.02) ^cd^
Ce	2.2 (0.2) ^de^	1.8 (0.2) ^b^	6.6 (0.5) ^cd^	66 (6) ^b^	0.33 (0.05) ^d^
IbP	5.1 (0.9) ^a^	4.2 (0.6) ^a^	6.5 (0.7) ^cd^	62 (6) ^c^	0.82 (0.13) ^a^
IbY	1.2 (0.3) ^g^	0.9 (0.3) ^d^	7.4 (0.6) ^ab^	77 (5) ^a^	0.16 (0.04) ^f^
OtW	1.74 (0.14) ^ef^	1.34 (0.08) ^c^	7.2 (0.4) ^ab^	72 (3) ^ab^	0.23 (0.03) ^e^
OtY	1.6 (0.2) ^fg^	1.3 (0.2) ^cd^	7.4 (0.3) ^a^	72 (3) ^ab^	0.22 (0.03) ^ef^
OtR	1.55 (0.06) ^fg^	1.17 (0.06) ^cd^	7.5 (0.5) ^abc^	70 (5) ^ab^	0.22 (0.02) ^ef^
Tt	2.9 (0.5) ^bc^	2.3 (0.5) ^b^	7.3 (0.5) ^abc^	74 (7) ^ab^	0.39 (0.08) ^bc^

Different letters in the columns represent the significant differences (*p <* 0.05) of the samples. F_s_—average specific force of structural ruptures; F_p_—average puncturing force; N_sr_—spatial frequency of structural ruptures; N_0_—number of peaks; W_c_—crispness work.

**Table 4 foods-13-00051-t004:** Color coordinates (L*, a*, b*, C* and h*) (mean values and standard deviations) and total color differences (ΔE) between the mixtures before extrusion and the 2G snacks.

Sample	L*	a*	b*	C*	h*	ΔE
Mixtures before extrusion						
Control	81.53 (0.04) ^Aa^	7.5 (0.2) ^Bfg^	42.7 (0.3) ^Ba^	43.4 (0.3) ^Ba^	80.1 (0.04) ^Aa^	-
Ax	79.5 (0.3) ^Ad^	5.3 (0.2) ^Bd^	27.8 (0.2) ^Bf^	28.3 (0.3) ^Be^	79.2 (0.3) ^Ae^	-
Ci	72.93 (0.02) ^Af^	6 (0.03) ^Be^	30.6 (0.2) ^Be^	31.2 (0.2) ^Bd^	79.01 (0.04) ^Acd^	-
Ce	83.67 (0.02) ^Ab^	4.13 (0.02) ^Bg^	23.87 (0.02) ^Bh^	24.22 (0.02) ^Bg^	80.18 (0.02) ^Ab^	-
IbP	73.5 (0.2) ^Ag^	7.17 (0.03) ^Bc^	27.2 (0.2) ^Bf^	28.14 (0.13) ^Be^	75.24 (0.02) ^Af^	-
IbY	74.0 (0.7) ^Af^	5.82 (0.14) ^Bde^	30.3 (0.2) ^Be^	30.8 (0.2) ^Bd^	79.1 (0.2) ^Ad^	-
OtW	80.89 (0.02) ^Ac^	6.3 (0.2) ^Bd^	37.9 (0.3) ^Bc^	38.4 (0.3) ^Bc^	80.58 (0.04) ^Ad^	-
OtY	79.46 (0.02) ^Ae^	6.53 (0.02) ^Bb^	39.64 (0.02) ^Bb^	40.17 (0.02) ^Bb^	80.64 (0.02) ^Ae^	-
OtR	79.24 (0.02) ^Ae^	3.4 (0.2) ^Bf^	26.7 (0.2) ^Bg^	26.9 (0.2) ^Bf^	82.7 (0.2) ^Abc^	-
Tt	67.21 (0.02) ^Ah^	8.86 (0.02) ^Ba^	32.8 (0.2) ^Bd^	34 (0.2) ^Bc^	74.89 (0.03) ^Ag^	-
2G Snacks						
Control	72.9 (0.2) ^Ba^	5.57 (0.03) ^Afg^	40.1 (0.2) ^Aa^	40.5 (0.2) ^Aa^	82.1 (0.2) ^Ba^	9.22 (0.06) ^f^
Ax	61.7 (0.2) ^Bd^	9.72 (0.14) ^Ad^	32.8 (0.8) ^Af^	34.2 (0.8) ^Ae^	73.47 (0.14) ^Be^	18.9 (0.2) ^c^
Ci	65.4 (0.2) ^Bf^	8.2 (0.2) ^Ae^	31.7 (0.02) ^Ae^	32.73 (0.03) ^Ad^	75.58 (0.13) ^Bcd^	7.97 (0.02) ^g^
Ce	66.7 (0.6) ^Bb^	8.3 (0.2) ^Ag^	32.3 (0.2) ^Ah^	33.31 (0.13) ^Ag^	75.7 (0.2) ^Bb^	19.35 (0.02) ^b^
IbP	63.7 (0.2) ^Bg^	9.3 (0.2) ^Ac^	32.9 (0.2) ^Af^	34.2 (0.2) ^Ae^	74.2 (0.2) ^Bf^	11.6 (0.2) ^d^
IbY	64.6 (0.3) ^Bf^	8.9 (0.4) ^Ade^	32.4 (0.3) ^Ae^	33.6 (0.4) ^Ad^	74.6 (0.6) ^Bd^	10.1 (0.6) ^e^
OtW	61.8 (0.4) ^Bc^	9.0 (0.9) ^Ad^	30 (0.4) ^Ac^	31.2 (0.5) ^Ac^	73.4 (0.9) ^Bd^	20.8 (0.2) ^a^
OtY	60.2 (0.2) ^Be^	10.69 (0.04) ^Ab^	32.47 (0.02) ^Ab^	34.19 (0.02) ^Ab^	71.8 (0.2) ^Be^	21 (0.02) ^a^
OtR	61 (0.3) ^Be^	9.7 (0.4) ^Af^	30.5 (0.7) ^Ag^	32 (0.7) ^Af^	72.4 (0.3) ^Bbc^	19.65 (0.02) ^b^
Tt	57.5 (0.4) ^Bh^	11.8 (0.2) ^Aa^	33.7 (0.2) ^Ad^	35.7 (0.2) ^Ac^	70.8 (0.2) ^Bg^	10.21 (0.02) ^e^

Different capital letters represent significant differences (*p* < 0.05) between processes, and lowercase letters represent significant differences (*p* < 0.05) between samples.

**Table 5 foods-13-00051-t005:** Total carotenoids (TC), total phenols (TP), and antioxidant capacity (AC) (mean values and standard deviations) of 2G snacks.

Sample	TC (mg_βcarotene_/100 g)	TP (mg_GAE_/100 g)	AC (mg_Trolox_/100 g)
C	1.75 (0.02) ^e^	111 (4) ^d^	0.27 (0.12) ^d^
Ax	1.75 (0.02) ^e^	123 (2) ^c^	4.6 (0.2) ^ab^
Ci	2.14 (0.06) ^c^	130.6 (1.2) ^a^	4.1 (0.4) ^b^
Ce	1.42 (0.03) ^f^	112 (3) ^d^	0.9 (0.2) ^d^
IbP	1.68 (0.02) ^e^	124.2 (1.3) ^c^	2.53 (0.14) ^c^
IbY	2.18 (0.02) ^c^	123.9 (1.3) ^c^	4.5 (0.3) ^ab^
OtW	1.96 (0.07) ^d^	131 (2) ^a^	4.58 (0.14) ^ab^
OtY	4.74 (0.04) ^a^	127.4 (0.8) ^b^	4.9 (0.2) ^a^
OtR	2.20 (0.04) ^c^	126 (2) ^bc^	1.9 (0.6) ^c^
Tt	3.76 (0.02) ^b^	124.2 (1.3) ^c^	4.8 (0.4) ^ab^

Different lowercase letters in the columns represent significant differences (*p <* 0.05) between samples.

## Data Availability

Data is contained within the article.

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
