# Peer review of "Characterization of Second-Generation Snacks Manufactured from Andean Tubers and Tuberous Root Flours"

_foods, 2023, doi:10.3390/foods13010051_

Round 1

Reviewer 1 Report

Comments and Suggestions for Authors

The document needs extensive modification. Introduction does not need to include the tables and figures you have; furthermore, one of the figures is also included in another article by the same authors.

Results and discussion also need extensive revision of the analysis, and the conclusions need to be shortened and focused on the objective of the manuscript

In addition, in this article, published a few weeks ago, the authors use similar materials. Some of the figures could be duplicated from this contribution. 

Foods 2023, 12(11), 2168; https://doi.org/10.3390/foods12112168 

Comments on the Quality of English Language

Requires English revision

Author Response

Dear Reviewer, thank you very much for your valuable remarks and suggestions. We considered them highly appropriate, and we included all of them in the present version of the manuscript (changes in the new document were underlined in grey). We express our appreciation for your time and effort in the preparation of comments and very valuable suggestions. Thank you very much.

Reviewer 2 Report

Comments and Suggestions for Authors

There is no logic in justifying the presented research. If there is a problem of hunger and malnutrition, people need to be offered cheap, nutritious main courses, not snacks. A snack is additional food between meals. The literature review should estimate the benefits to society resulting from the introduction of the proposed snacks to the market

It is worth discussing the chemical composition of individual raw materials in the review. Various product properties will depend on this. The question also needs to be answered whether the proposed raw materials are edible for humans. It is also a pity that no sensory evaluation was carried out, so it was not assessed whether the proposed products would be accepted by consumers

The conclusion contains statements that have no reference to the scope of the research conducted.

However, the authors should recommend the products they consider most worth recommending, of course based on the results obtained.

detailed comments

Line 22 “……flour showed better characteristics…. “ better compared to what?

Table 1 This table does not say much and is perceived ambiguously. e.g. it may mean that fruit is very cheap, twice as cheap as in the world, or that you don't buy much of it

Line 100 to 104 markings of samples - provide them in the table, it will be easier to see. the composition of the snacks is unclear. Were they always made of two ingredients, i.e. corn flour and one plant?

Line 136-7 It is worth briefly describing these methods

Figure caption 4. Please refer to the explanation of the symbols (paragraph or proposed table) and do not repeat it

I propose to move paragraph line 195-5 together with table 2 and place it in line 174, immediately after subtitle 3.1. Characterization of 2G Snacks.

  Line 238 – 240 In this sentence, the authors refer to rheological rather than chemical features

Line 265: The authors did not measure pore size, so they cannot say anything about pore size

Line 271,

line291. Reference to table 2, not 3

Table 3 should be placed in subsection 3.1.6. Texture properties

Line 366 It should be Figure 5, not figure 5

Conclusion

Line 434-440 The content of bioactive compounds before extrusion was not compared, so the statement about losses during the process is groundless. Similarly, costs were not analyzed!

The content of starch, fiber and proteins was not tested - so it cannot be commented on

Author Response

(The authors gave the same response as above.)

Reviewer 3 Report

Comments and Suggestions for Authors

The article entitled " Characterization of second-generation snacks manufactured from Andean tubers and tuberous root flours’ appears comprehensive in its scope and objectives, here are some points of improvement:

1. Clearly define "second-generation (2G) snacks." Condense the list of roots, e.g., "Andean roots like zanahoria blanca, achira, etc. contain..." Highlight the main findings prominently.

2. Provide detailed procedures for replication. Use clear headings/subheadings for each method subsection. Ensure consistent use of units, e.g., consistently use Celsius for temperatures. Explain terms like "hygroscopicity" and "bulk density."

3. Clarify sampling methods for Andean roots, specifying the number of samples.

4. Give detailed information on equipment, like the Brabender laboratory extruder. Elaborate on the selection of software and tests used.

5. Present data, like water content, in tables or graphs.

6. Define all acronyms upon first use, such as "2G" and "3G."

7. Integrate and provide context for citations (e.g., [20], [21,22], [23]). Delve deeper into interpreting the data.

8. Explain the significance of findings regarding 2G snacks.

9. Discuss causes for observed differences in samples.

10. Discuss implications for the food industry and the real-world applications of 2G snacks.

11. Ensure proper citation formatting.

12. Concisely summarize key findings and their broader significance.

Enhance the article's clarity, flow, and readability to amplify its impact.

Comments on the Quality of English Language

Minor editing of English language required

Author Response

(The authors gave the same response as above.)

Round 2

Reviewer 1 Report

Comments and Suggestions for Authors

My concern is still the same, as the present document could be included in the previous paper

Comments on the Quality of English Language

moderate revision

Author Response

Dear Reviewer, thank you very much for your valuable remarks and suggestions.

Reviewer 2 Report

Comments and Suggestions for Authors

Accept in present form

Author Response

Dear Reviewer, we express our appreciation for your time and effort in the preparation of comments and very valuable suggestions.

Thank you very much.